# Cardiopulmonary Fitness of Preschoolers with Congenital Heart Disease: An Observational Study

**DOI:** 10.3390/metabo13010118

**Published:** 2023-01-11

**Authors:** Yen-Yu Chen, Chung-Lan Kao, Sheng-Hui Tuan, Ko-Long Lin

**Affiliations:** 1Department of Physical Medicine and Rehabilitation, Taipei Veterans General Hospital, Taipei 11221, Taiwan; 2School of Medicine, National Yang Ming Chiao Tung University, Taipei 11221, Taiwan; 3Department of Physical Therapy and Assistive Technology, National Yang Ming Chiao Tung University, Taipei 11221, Taiwan; 4Center for Intelligent Drug Systems and Smart Bio-Devices (IDS2B), National Yang Ming Chiao Tung University, Hsinchu 300093, Taiwan; 5Institute of Allied Health Sciences, College of Medicine, National Cheng Kung University, No. 1, University Rd., Tainan City 70101, Taiwan; 6Department of Rehabilitation Medicine, Cishan Hospital, Ministry of Health and Welfare, No. 60, Zhongxue Rd., Cishan District, Kaohsiung 84247, Taiwan; 7Department of Physical Medicine and Rehabilitation, Kaohsiung Veteran General Hospital, Kaohsiung 81362, Taiwan; 8Department of Physical Medicine and Rehabilitation, School of Medicine, College of Medicine, Kaohsiung Medical University, Kaohsiung 80756, Taiwan

**Keywords:** preschoolers, congenital heart disease, exercise testing, cardiopulmonary fitness, sex difference, fat-free mass index

## Abstract

With advancements in cardiopulmonary rehabilitation over the past few decades, the survival rate of patients with congenital heart disease (CHD) has increased. However, the Cardiopulmonary fitness (CPF) of these patients is poor. Here, we aimed to investigate CPF in preschoolers with CHD (aged 4 to 6 years) using cardiopulmonary exercise testing. We retrospectively compared 102 healthy preschoolers with 80 preschoolers with CHD. The latter had lower peak oxygen consumption, oxygen consumption at anaerobic threshold and metabolic equivalent at anaerobic threshold. The same result was observed in boys with CHD, but not in girls, when sex was sub-analyzed. Considering the body composition, children with CHD had a lower fat-free mass index (FFMI) than their healthy peers. Healthy preschoolers with a normal body mass index (BMI) had higher anaerobic threshold and peak metabolic equivalent values than overweight or underweight children. This was categorized under the BMI reference of the Ministry of Health and Welfare in Taiwan. In conclusion, the CPF difference between the CHD and healthy groups was identified as early as in preschool age, and better CPF in healthy preschoolers within the normal BMI range suggests the importance of weight control in young children.

## 1. Introduction

Congenital heart disease (CHD), defined as any structural abnormalities of the heart or great vessels that is presented at birth, is a common birth defect worldwide. CHD affects 0.8–1% of live births [1]. The prevalence of CHD in China is approximately 9 per 1000 live births [2]. In the United States, CHD affects around 40,000 births annually [3]. There are two common groups of CHD, namely acyanotic (“pink”) and cyanotic (“blue”) CHDs. Acyanotic heart diseases include atrial septal defect (ASD), ventricular septal defect (VSD), atrioventricular septal defect, patent ductus arteriosus, and aortic and pulmonary valve stenosis. Cyanotic heart diseases include tetralogy of Fallot (TOF), transposition of great arteries and truncus arteriosus. VSD, one of the most common lesion types, accounts for 50–60% of CHD cases [4].

With the development of prenatal screening techniques and medical and surgical advancements, the mortality rate of CHD has declined globally [1]. Corrective surgeries have increased the survival rates of children with CHD; however, comorbidities persist in other systems, such as pulmonary, neurodevelopmental and psychiatric problems [5,6,7]. The exercise capacity of CHD survivors is an emerging issue. Whether children with CHD are able to sustain similar levels of intensity in physical activities with peers is a question frequently raised by families and health practitioners [8,9]. A previous study examined the symptoms, parental health concerns, and activity levels of pre-school children using questionnaires and accelerometers. Children with CHD had comparable physical activity habits and screen and activity time amounts as age- and sex-matched controls [9]. One Canadian study using accelerometers and questionnaires reported that children with CHD spent similar amounts of time engaged in moderate to vigorous physical activity [10]. Conversely, parental overprotection might prevent children with CHD from performing the same activities as other children of the same age [11].

The aforementioned studies focused on physical activity, defined by the World Health Organization as any body movement requiring energy expenditure. To enable a more precise and objective evaluation, cardiopulmonary exercise testing (CPET) is the golden standard for measuring cardiopulmonary fitness (CPF) and the functional capabilities of the heart, lungs and muscles relative to the demands of specific exercise routines. This measurement has been used in several patients with cardiopulmonary diseases [12]. In the CHD population, previous studies evaluated CPF with conflicting results. However, most studies focused on adults and adolescents [13,14,15]. To our knowledge, no study has evaluated CPF in preschoolers with CHD. Moreover, differences in body composition and sex may be crucial factors that influence exercise capacity. Our previous studies revealed that preschoolers with a higher fat-free mass index (FFMI) had significantly higher aerobic fitness levels, and girls had significantly better CPF during CPET [16,17,18]. Thus, we aimed to evaluate the exercise capacity of preschool children with CHD using CPET and assess whether sex or excess body fat influences CPET performance.

## 2. Materials and Methods

### 2.1. Subjects’ Characteristics

This retrospective cross-sectional study was conducted in a single medical center located in southern Taiwan from February 2018 to February 2022. We enrolled two groups of preschoolers, one with known CHD and a control group that included age-, sex-, and body mass index (BMI)-matched children referred to the cardiopulmonary rehabilitation outpatient clinic during the same period due to dyspnea on exertion or chest pain. The preschoolers in the control group had no previous history of CHD or diagnosis with any cardiopulmonary disease after a series of examinations, including echocardiography and 12-lead electrocardiography. All participants underwent CPET at Kaohsiung Veterans General Hospital, Taiwan.

The inclusion criteria were as follows:Aged 4 to 6 years;Understood the CPET steps; andNo signs of acute infection or fever 3 days prior to the CPET.

The exclusion criteria were as follows:Acquired heart diseases such as Kawasaki disease;Failure to complete CPET due to muscle fatigue;Known concurrent pulmonary disease; andRefusal to participate.

All preschoolers underwent a body composition examination prior to the CPET. All data of the included preschoolers were reviewed by a physiatrist (K-L LIN, who had more than 15 years’ experience administering CPET). The parents of all participants provided informed consent prior to the body composition measurement and CPET. This study was approved by the Institutional Review Board of Kaohsiung Veterans General Hospital (No. VGHKS17-CT11-11) and conducted in accordance with the Helsinki Declaration.

### 2.2. Anthropometry and Body Composition

Zeus 9.9 PLUS (Jawon Medical Co., Ltd., Kungsang Bukdo, Korea) was used to perform the body composition analysis via vector bioelectrical impedance analysis (VBIA). The tetrapolar electrode method (in which electrodes were attached to the bilateral hands, soles, and ankles) was used. After entering the participants’ basic data, including height, weight, age and sex, the analyzer calculated the body mass index (BMI) and fat mass index (FMI). The participants were categorized into four groups (underweight, normal, overweight, and obese) according to the age- and sex-specific reference BMI values suggested by the Ministry of Health and Welfare in Taiwan [19]. BMI was defined as body weight (kg) divided by height squared (m^2^), FMI was defined as fat mass (FM) (kg) divided by height squared (m^2^), and FFMI was defined as the difference between the body weight and FM divided by height squared (m^2^).

### 2.3. Cardiopulmonary Exercise Testing (CPET)

A treadmill was used for the CPET. According to previous studies and the 11th edition of the American College of Sports Medicine (ACSM) guidelines, the treadmill is useful in a wide range of patient ages and sizes compared to the cycle ergometer, which requires additional attention and volitional effort to maintain the cadence [20]. A treadmill can be used safely from 3 years of age [21,22], and yields higher peak oxygen uptake (VO_2peak_) and maximal heart rate [12].

The test was supervised by an experienced physiatrist (K-L LIN), and all children and parents were familiar with the CPET steps before starting. We performed graded symptom-limited CPET using the ramped Bruce protocol with a flow module, gas analyzer, and electrocardiographic monitor (Metamax 3B, Cortex Biophysik GmbH Co., Leipzig, Germany). The test was terminated when the preschoolers demonstrated subjective unbearable symptoms; could no longer continue; or attained maximal effort as indicated by the ACSM criteria. To be considered as maximal effort, the preschoolers must have achieved at least 1 of the following criteria: (1) peak respiratory exchange ratio (RER) > 1.1, and (2) peak heart rate (HR) > 85% of the age-predicted maximum [12]. Oxygen uptake (VO_2_) and carbon dioxide production (VCO_2_) were measured using the breath-by-breath method. Blood pressure and heart rate were also measured. Metabolic equivalent (MET) is considered equivalent to 3.5 mL of oxygen per kilogram of body mass per minute [23]. Peak MET was defined as the maximal MET throughout CPET. Anaerobic threshold (AT) was determined using the ventilatory equivalents for O_2_ and CO_2_ methods [24].

### 2.4. Statistical Analysis

SPSS for Windows version 21.0 (IBM Corp., Armonk, NY, USA) was used for the analyses. Continuous data are presented as mean ± standard deviation and categorical variables as absolute numbers or percentages. We checked the normality and homogeneity of variances and outliers before the analyses. All the comparisons of healthy controls and preschoolers with CHD were performed using an independent-*t* test. We used the chi-Squared test or Fisher’s exact test to compare the percentage of excessive adiposity between the healthy control group and preschoolers with CHD. One-way analysis of variance was used for intragroup comparisons of subjects with different FM values, while an independent-*t* test was used for intergroup comparisons of the healthy and CHD groups. Post-hoc tests were performed to analyze the variance among different BMI groups. Statistical significance was set at *p* < 0.05.

## 3. Results

### 3.1. Patients’ Demographic Characteristics

After the exclusion of outliers in the control group and those for whom data were missing, 182 preschoolers were included in the study. Among them, 102 children were in the control group (56 boys, 46 girls) and 80 were in the CHD group (48 boys, 32 girls). The most common CHD types were ASD (*n* = 17), VSD (*n* = 18) and TOF (*n* = 18). Table 1 presents the preschoolers’ baseline demographic characteristics. The mean ages of the total healthy group, boys and girls were 5.65 ± 0.59, 5.68 ± 0.61 and 5.61 ± 0.58 years, respectively. The mean ages of the total CHD group, boys, and girls were 5.78 ± 0.4, 5.88 ± 0.33 and 5.63 ± 0.61, respectively. The mean FFMI of the control group was significantly higher than that of the CHD group (*p* = 0.041). There were no significant intergroup differences in terms of height, weight, BMI, body fat, FM, FMI and FFM. However, we observed a tendency, approaching statistical significance (*p* = 0.061), for the mean BMI of the CHD group to be lower than that of the healthy group for both boys and girls.

### 3.2. Comparisons of Cardiopulmonary Fitness between CHD and Control Groups

Table 2 compares the CPF levels of the CHD and healthy groups by BMI. The VO_2peak_ and AT VO_2_ were recorded as absolute oxygen consumption (mL/min), while peak MET and AT represented the relative aerobic capacity (mL/min/kg). The percentage of measured VO_2peak_ to the predicted value was calculated and compared with the healthy subjects. A value of 100% was considered normal, while a value of ≥80% was considered desirable. Peak rate pressure product (RPP), expressed as resting heart rate (in bpm) multiplied by systolic blood pressure (in mmHg), represents the myocardial oxygen uptake during the CPET [12].

Irrespective of sex, the mean peak RER value was 1.14 in both groups (*p* = 0.815), suggesting that maximal oxygen exercise efforts were reached. Metabolic equivalent at anaerobic threshold (AT MET), AT VO_2_ and VO_2peak_ were significantly higher in the control group versus CHD group (*p* = 0.015, 0.005 and 0.048, respectively). There were no significant intergroup differences in peak MET, PD, peak RPP, or heart rate recovery. Similar results were observed in the boys but not girls.

**Table 1 metabolites-13-00118-t001:** Baseline characteristics of all recruited preschool children.

	Age (years)	Height (cm)	Weight (kg)	BMI (kg/m^2^)	Body Fat (%)	U (%)	N (%)	O (%)	F (%)	FM (kg)	FMI (kg/m^2^)	FFM (kg)	FFMI (kg/m^2^)
Control-total (N = 102)	5.6 ± 0.6	118.5 ± 7.2	22.0 ± 5.5	15.6 ± 3.1	13.6 ± 7.5	28.4	56.9	5.9	7.9	3.4 ± 2.6	2.4 ± 1.8	20.0 ± 4.8	14.0 ± 2.1
CHD-total (N = 80)	5.8 ± 0.5	118.7 ± 9.2	21.2 ± 6.0	14.8 ± 2.3	12.7 ± 5.3	31.3	61.3	5.0	1.3	3.0 ± 2.3	2.0 ± 1.1	19.2 ± 5.0	13.2 ± 1.9
*p* value ^a^	0.108	0.836	0.334	0.061	0.447	0.363 ^b^				0.454	0.155	0.417	0.041 *
Control-boys (N = 56)	5.7 ± 0.6	119.6 ± 8.9	23.1 ± 6.5	16.0 ± 3.5	11.6 ± 6.8	21.4	60.7	5.4	10.7	3.1 ± 2.8	2.1 ± 2.0	21.6 ± 5.9	14.7 ± 2.4
CHD-boys (N = 48)	5.9 ± 0.3	118.4 ± 6.7	21.1 ± 5.4	14.9 ± 2.5	10.9 ± 5.1	35.4	58.3	4.2	2.1	2.7 ± 2.1	1.8 ± 1.2	19.9 ± 4.9	13.7 ± 2.1
*p* value ^a^	0.148	0.465	0.092	0.060	0.661	0.302 ^b^				0.540	0.382	0.238	0.079
Control-girls (N = 46)	5.6 ± 0.6	117.2 ± 4.0	20.7 ± 3.7	15.0 ± 2.3	15.9 ± 7.7	37.0	52.2	6.5	4.3	3.7 ± 2.4	2.7 ± 1.7	18.1 ± 1.7	13.0 ± 1.1
CHD-girls (N = 32)	5.6 ± 0.6	119.2 ± 10.2	21.3 ± 7.0	14.8 ± 2.1	14.5 ± 4.95	25.0	65.6	6.3	0	3.4 ± 2.5	2.2 ± 1.0	18.6 ± 5.1	12.7 ± 1.5
*p* value ^a^	0.905	0.295	0.604	0.585	0.400	0.419 ^b^				0.612	0.226	0.646	0.317

BMI: body mass index; U (%), percentage of underweight subjects; N (%), percentage of normal weight subjects; O (%), percentage of overweight subjects; F (%), percentage of obese subjects; FM, fat mass; FMI, fat mass index; FFM, fat-free mass; FFMI, fat-free mass index. ^a^ All the comparisons between normal controls and preschoolers with congenital heart disease were carried out by independent-*t* test except *p* values marked with ^b^, which were analyzed by Fisher’s exact test for the comparison percentage of excessive adiposity between normal controls and preschoolers with congenital heart disease. * *p* value < 0.05

**Table 2 metabolites-13-00118-t002:** Comparisons of cardiopulmonary fitness between subjects with underweight, normal, and overweight body mass index.

	AT MET			ATVO_2_			Peak MET			PeakVO_2_			Peak PD			Peak RPP		
	Control	CHD	*p* Value ^b^	Control	CHD	*p* Value ^b^	Control	CHD	*p* Value ^b^	Control	CHD	*p* Value ^b^	Control	CHD	*p* Value ^b^	Control	CHD	*p* Value ^b^
**Boys and Girls**																		
**Total** (N = 102;80)	8.1 ± 1.5	7.5 ± 1.6	0.015 *	616.1 ± 156.4	550.3 ± 152.7	0.005 *	10.8 ± 1.9	10.4 ± 2.0	0.160	828.9 ± 224.1	763.7 ± 212.5	0.048 *	80.5 ± 17.7	76.0 ± 21.3	0.124	29,568.2 ± 33,388.5	25,025.8 ± 5672.6	0.231
**U** (N = 29;25)	7.9 ± 1.4	7.6 ± 1.9	0.643	508.9 ± 106.9	488.1 ± 157.6	0.569	10.6 ± 1.7	10.6 ± 2.3	0.962	686.0 ± 128.7	677.4 ± 193.0	0.190	82.8 ± 17.9	77.0 ± 21.6	0.282	27,350.8 ± 7428.0	23,451.0 ± 5790.1	0.038 *
**N** (N = 57;48)	8.4 ± 1.5	7.5 ± 1.4	0.005 *	631.8 ± 135.6	568.5 ± 132.6	0.018 *	11.2 ± 2.0	10.5 ± 1.9	0.065	850.0 ± 225.0	790.8 ± 208.6	0.164	81.4 ± 17.6	76.7 ± 21.5	0.211	31,635.4 ± 44,069.3	25,524.8 ± 5551.1	0.338
**O** (N = 16;7)	7.3 ± 1.0	6.5 ± 1.8	0.436	764.0 ± 173.6	663.4 ± 202.9	0.312	9.8 ± 1.2	8.9 ± 2.0	0.458	1023.6 ± 195.7	902.0 ± 222.5	0.161	72.2 ± 16.7	66.4 ± 18.2	0.815	26,000.2 ± 6542.2	27,512.5 ± 5422.9	0.312
*p* value ^a^	0.028 *^,d^	0.289		<0.001 *^,c,d,e^	0.015 *^,c^		0.031 *^,d^	0.157		<0.001 *^,c,d,e^	0.022 *		0.141	0.521		0.775	0.178	
**Boys**																		
**Total** (N = 56;48)	8.2 ± 1.6	7.3 ± 1.6	0.009 *	651.6 ± 165.2	538.6 ± 152.8	0.001 *	10.9 ± 2.2	10.3 ± 2.2	0.178	887.6 ± 253.8	757.3 ± 206.2	0.010 *	70.5 ± 13.9	65.9 ± 19.0	0.154	26,262.3 ± 5753.7	24,311.7 ± 5905.7	0.093
**U** (N = 12;17)	7.8 ± 2.0	7.3 ± 1.6	0.807	517.2 ± 140.2	463.1 ± 126.1	0.308	10.4 ± 2.2	10.5 ± 2.2	0.947	690.7 ± 165.9	663.6 ± 187.0	0.658	66.7 ± 12.5	68.3 ± 15.5	0.707	27,522.3 ± 7104.7	21,944.5 ± 5385.2	0.057
**N** (N = 33;27)	8.6 ± 1.4	7.4 ± 1.6	0.004 *	660.7 ± 124.2	557.4 ± 131.1	0.003 *	11.5 ± 2.3	10.5 ± 2.2	0.081	892.3 ± 251.7	781.6 ± 186.0	0.058	73.5 ± 14.5	65.6 ± 21.1	0.086	25,627.6 ± 4556.6	25,271.4 ± 6042.8	0.794
**O** (N = 11;4)	7.2 ± 1.0	6.5 ± 2.2	0.866	782.9 ± 203.3	797.4 ± 180.5	0.735	9.7 ± 0.9	8.6 ± 2.6	0.865	1051.8 ± 223.1	1062.3 ± 172.9	0.866	64.9 ± 11.7	54.9 ± 16.8	0.612	26,844.7 ± 7690.5	28,768.3 ± 2080.7	0.612
*p* value ^a^	0.029 *^,d^	0.660		<0.001 *^,c,e^	0.001 *^,c,d^		0.044 *^,c,e^	0.400		0.002 *^,c,e^	0.003 *^,c,d^		0.129	0.534		0.592	0.072	
**Girls**																		
**Total** (N = 46;32)	7.9 ± 1.3	7.7 ± 1.6	0.576	573.7 ± 135.1	567.4 ± 153.3	0.847	10.6 ± 1.5	567.4 ± 153.3	0.612	769.6 ± 163.6	773.2 ± 224.5	0.935	92.6 ± 14.1	91.2 ± 14.5	0.669	33,521.0 ± 49,074.0	26,096.9 ± 5210.6	0.398
**U** (N = 17;8)	7.9 ± 1.0	8.3 ± 2.3	0.448	503.0 ± 71.0	541.3 ± 209.9	0.771	10.7 ± 1.2	541.3 ± 209.9	0.793	682.7 ± 100.0	706.7 ± 215.2	0.683	94.2 ± 11.0	95.4 ± 22.0	0.838	27,229.8 ± 7862.3	26,652.3 ± 5606.6	0.907
**N** (N = 24;21)	8.0 ± 15.6	7.7 ± 1.7	0.419	592.0 ± 143.0	582.7 ± 136.5	0.825	10.7 ± 1.6	529.7 ± 136.5	0.545	790.0 ± 167.7	803.0 ± 239.6	0.831	92.6 ± 15.7	91.5 ± 10.4	0.835	39,896.0 ± 67,651.5	25,862.6 ± 4944.8	0.349
**O** (N = 5;3)	7.5 ± 1.3	6.5 ± 1.7	0.456	726.2 ± 99.2	529.4 ± 128.1	0.101	9.9 ± 1.9	529.4 ± 239.1	0.653	967.2 ± 126.9	741.7 ± 129.8	0.025 *	86.8 ± 16.5	77.9 ± 12.0	0.453	24,311.2 ± 3373.1	26,256.7 ± 8028.6	0.456
*p* value ^a^	0.671	0.254		0.002 *^,c^	0.744		0.544	0.283		0.001 *^,c,d^	0.583		0.597	0.205		0.660	0.938	

CHD, congenital heart disease; U, underweight, N, normal body mass index, O, overweight and obesity; AT MET, metabolic equivalent at anaerobic threshold; Peak MET, peak metabolic equivalent during exercise testing; AT VO_2_, oxygen consumption at anaerobic threshold; Peak VO_2_, peak oxygen consumption during exercise testing; peak PD, percentage of measured peak oxygen consumption to predicted value; RPP, rate pressure product.* *p* value < 0.05, ^a^ intra-group comparisons between subjects with different body adiposity by one-way analysis of variance, ^b^ inter-group comparisons between normal control and preschooler with congenital heart disease by independent-*t* test. Post-hoc analysis via the Bonferroni test found significant difference between ^c^ overweight and underweight, ^d^ overweight and normal, ^e^ normal and underweight.

### 3.3. Comparisons of Cardiopulmonary Fitness among Different BMIs

Table 2 compares CPF levels by BMI groups (underweight, normal, and overweight/obese) defined by the Ministry of Health and Welfare in Taiwan [19]. Among the healthy preschoolers, the normal BMI group has a significantly higher AT (*p* = 0.028) and peak MET (*p* = 0.03) than the underweight and overweight/obese groups. This result was not observed in children with CHD. Similarly, a subgroup analysis by sex showed that healthy boys with a normal BMI had a significantly higher AT (*p* = 0.029) and peak MET (*p* = 0.044) than healthy boys in the other BMI groups, whereas healthy girls with a normal BMI did not.

## 4. Discussion

This observational study revealed that preschoolers with CHD had lower VO_2peak_, ATVO_2_ and AT MET values than their healthy peers. Considering body composition, preschoolers with CHD had a lower FFMI than their healthy counterparts, while healthy preschoolers with a normal BMI had higher AT and peak MET values than those classified as underweight or overweight/obese.

In accordance with previous research, our study showed that children with CHD had poorer CPF as assessed by CPET. Villaseca-Rojas et al. reviewed 19 articles and stated that children aged 5–17 years with CHD have worse VO_2peak_, maximum workload, ventilatory equivalent for CO_2_ slope, O_2_ pulse, and maximal heart rate values. This difference was even greater among adolescents [25]. Our study findings support Villaseca-Rojas’s findings that younger children with CHD had lower VO_2peak_, AT VO_2_ and AT MET values. There are several possible factors related to the poorer exercise capacity among preschoolers with CHD: congenital structural abnormalities, hemodynamics differences in individuals with CHD, late complications, and the influence of surgery (such as a nidus causing arrhythmia) [26]. Moreover, parental overprotection and physician caution might prevent children with CHD from engaging in the same physical activity levels as their peers [11]. Since the VO_2peak_ data in this study were not adjusted for body weight, the intergroup differences in body weight may have influenced the results. MET might be a better marker of CPF than VO_2_. The lower MET at AT in preschoolers with CHD suggests poorer exercise capacity [24].

In our study, the FFMI was higher in the control versus CHD group. The mean BMI in preschoolers with CHD was insignificantly lower than that of the healthy controls. Previous studies reported that the risk of obesity in children with CHD increased over time, especially in patients with mild CHD [27,28]. Adipose composition varies among countries and age groups. A Danish national study analyzing the prevalence of underweight or obesity in 1–15-year-old children summarized that the median BMI was slightly lower for CHD individuals than the general population for both sexes [29]. Interestingly, a study conducted in Taiwan reported a shift in BMI from underweight to overweight with increasing age. The study showed a higher underweight prevalence in first graders versus a higher proportion of overweight/obesity in adolescents, especially in males [30]. Thus, being underweight at a young age should not be an excuse for a sedentary life; rather, encouraging preschoolers with CHD to participate in appropriate physical activity is necessary for their future health.

In our previous work [16], preschoolers with excess body adipose tissue according to BMI and FFMI classifications showed a poorer ability to achieve maximal effort during CPET, as evidenced by the lower oxygen uptake efficiency slope. However, in contrast to previous results that showed no significant difference in AT or peak MET, the current study showed higher AT and peak MET values in the normal BMI group. On the other hand, higher AT and peak MET in the normal BMI group were not seen in the CHD preschoolers. Poor cooperation and motivation among preschoolers and the relatively small number of participants may have contributed to the different results.

Finally, we observed that preschoolers with CHD had a lower VO_2peak_, AT VO_2_ and AT MET than healthy controls. Similar results were found among CHD boys but not in girls with CHD. Our previous study [31] suggested the discrepancy in peak VO2 between sexes was noted as early as at preschool age. This study showed similar results in preschoolers with CHD. Consistent with our research on children with TOF [18] and Kawasaki disease [32], the affected girls had better CPF than the affected boys. Self-efficacy, cultural differences, and gender stereotypes in sports participation may have contributed to this finding. Self-efficacy is believed to be reduced in patients with heart disease. A cross-sectional study using objective assessments, questionnaires, and interviews suggested that the environment and family influence the positive perception of activity participations [33]. A patriarchal culture affecting some families, with overprotection of boys and social expectation differences, might have led to those results.

The main strength of the current study is that it is the first to analyze CPF in preschoolers with CHD using CPET. Nevertheless, it had some limitations. First, this was a single-center study of participants from southern Taiwan only, and regional differences should be considered. Moreover, although our investigation of a relatively young population made our study unique, we could not confirm the representativeness of our recruited participants because of a lack of comparable reference values. Second, the study included a relatively small number of participants compared to some national database studies. Third, the control group included preschoolers who had symptoms such as palpitations and dyspnea, but no evidence of cardiopulmonary disease. Some problems might not have been found with the current medical technology. In addition, we did not analyze the CPET results by CHD types or severities. Finally, the theory of sex-based differences and cultural perspectives has not been proven by qualitative surveys such as pedometers or questionnaires.

## 5. Conclusions

To the best of our knowledge, this study is the first to evaluate CPF in children with CHD aged 4–6 using CPET. The CPF difference between healthy and CHD patients was identified as early as preschool age. Moreover, better CPF in healthy preschoolers within the normal BMI range suggests the importance of weight control in young children. These results suggest that promoting an active lifestyle to prevent future functional deterioration and to reduce the gap between CHD and healthy children is essential. Future larger and prospective investigations of the exercise capacity of healthy versus diseased preschoolers are warranted.

## Data Availability

Data sharing statement: Individual participant data that underlie the results reported in this article, after deidentification, might be shared. Proposals should be directed to kllin@vghks.gov.tw. for application.

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
