# Peer review of "Cardiopulmonary Fitness of Preschoolers with Congenital Heart Disease: An Observational Study"

_metabolites, 2023, doi:10.3390/metabo13010118_

Round 1

Reviewer 2 Report

the FFMI is body weight - FM/ height?

the tables are very difficult to read in this layout. There is a comparion between control and chd, a comparison between under, normal and overweight and between boy's and girls and a comparisons of combinations groups.

3.3 states that N groep has higher AT and peak MET in control compared to U en O group, this was not seen in CHD. the subgroup analysis described in last centences I don't understand.

a possible reason for none effect op BMI in CHD on exercise capacity is not given. Why in controle group exercise was beter in normal bmi compared to U or O group, which was not seen in CHD groups?

Round 2

Reviewer 1 Report

Thanks to the authors for the substantial improvements of the manuscript 

Reviewer 2 Report

in table 1 and 2 still to cluttered. 2 digits after the comma is not necessary.

the conlusion in abstract doesnot correspond with the conclusion in the article. in the abstract is claimed " Promoting an active lifestyle to prevent future functional deterioration and to reduce the gap between CHD and healthy children is essential " but this was not investigated.

Round 3

Reviewer 2 Report

all changes are made and current form is fine.